# Analysis of the Spatial Distribution and Deformation Types of Active Landslides in the Upper Jinsha River, China, Using Integrated Remote Sensing Technologies

Shengsen Zhou [1], Baolin Chen [1,2], Huiyan Lu [1], Yunfeng Shan [1], Zhigang Li [1], Pengfei Li [3], Xiong Cao [4] and Weile Li [1,5,*]

1  State Key Laboratory of Geohazard Prevention and Geoenvironment Protection, Chengdu University of Technology, Chengdu 610059, China
2  Xi'an Division of Surviving and Mapping, Xi'an 710054, China
3  Guiyang Engineering Corporation Limited of Power China, Guiyang 550081, China
4  Southwest Branch of China Petroleum Engineering and Construction Co., Ltd., Chengdu 610041, China
5  Laboratory of Landslide Risk Early-Warning and Control, Chengdu 610059, China
*  Correspondence: liweile08@mail.cdut.edu.cn

**Abstract:** The Upper Jinsha River (UJSR) has great water resource potential, but large-scale active landslides hinder water resource development and utilization. It is necessary to understand the spatial distribution and deformation trend of active landslides in the UJSR. In areas of high elevations, steep terrain or otherwise inaccessible to humans, extensive landslide studies remain challenging using traditional geological surveys and monitoring equipment. Stacking interferometry synthetic aperture radar (stacking-InSAR) technology, optical satellite images and unmanned aerial vehicle (UAV) photography are applied to landslide identification. Small baseline subset interferometry synthetic aperture radar (SBAS-InSAR) was used to obtain time-series deformation curves of samples to reveal the deformation types of active landslides. A total of 246 active landslides were identified within the study area, of which 207 were concentrated in three zones (zones I, II and III). Among the 31 landslides chosen as research samples, six were linear-type landslides, three were upward concave-type landslides, 10 were downward concave-type landslides, and 12 were step-type landslides based on the curve morphology. The results can aid in monitoring and early-warning systems for active landslides within the UJSR and provide insights for future studies on active landslides within the basin.

**Keywords:** the Upper Jinsha River; active landslide; integrated remote sensing; spatial distribution; time-series deformation curves

## 1. Introduction

Since the Middle Pleistocene, with the uplift of the Qinghai-Tibet Plateau (QTP), downcutting of rivers, and concentrated high rainfall, the peripheral area of the plateau has become an area with some of the most serious landslide disasters in China [1,2]. The Jinsha River (JSR) is located on the southeastern edge of the QTP, and its upstream section is a hotspot for the development of hydropower projects in China [3]. At present, the layout of planned and under-construction hydropower stations is characterized as "one reservoir and thirteen stages" [4,5]. Hydroelectricity is a viable source of environmentally friendly and high-quality energy. Nevertheless, the advancement of hydropower is limited by active landslides [6,7]. In the Upper JSR (UJSR), a number of historical landslide-flood events have taken place and caused significant damage [8]. For instance, two high-locality landslides occurred in the UJSR near the village of Baige in Jiangda County, Tibet, on 11 October 2018 and 3 November 2018. Large rock masses blocked the JSR, which subsequently impounded a 290 million $m^3$ lake, affecting more than 20,000 residents and causing enormous economic losses and social impacts [9].

After the Baige landslide blocked the river, much research has been carried out on active landslides in the UJSR. Numerous studies have concentrated on large landslides that pose a threat to the UJSR [10]. The failure mechanism and deformation trends of the Woda landslide [11,12], Xiongba landslide [13], Xiaomojiu landslide [14], Wangdalong landslide [15] and "unstable blocks" of the Baige landslide [16–18] were analyzed through field examination and numerical simulation. Alternatively, some scholars have been able to detect active landslides in the UJSR by employing remote sensing technology, including high-resolution optical satellite images, InSAR, and UAV photography [19–21]. The above studies represent good references for the study of active landslides in the UJSR, but the systematic research results on the deformation types of active landslides are relatively lacking.

A number of surface deformation features will appear during landslide development. These deformation signals can be detected and monitored using certain types of technology and equipment, including total stations [22], borehole inclinometers [23], time domain reflectometry (TDR) [24], optical fiber sensing technology [25], global navigation satellite systems (GNSS) [26], RGB-D sensors [27] and others. Then, the collected data are plotted on time-series deformation curves to observe the displacement trend of the landslide. Miao et al. [28] classified landslides into four categories based on the time-series deformation curves: steady-type landslides, accelerated-type landslides, step-like landslides, and convergent-type landslides. Xiong et al. [16] and Carlà, T. et al. [29] detected clear trends of accelerating displacement prior to the failure of the Baige landslide and the Xinmo landside. Currently, the most common method for collecting high-accuracy and real-time displacement data is to install GNSS in the high-risk area. However, the surface-installed equipment is limited to measuring the displacement of a single landslide. Furthermore, some of these systems are costly, and the installation is challenging in high-elevation valley locations [26,30]. Satellite-based interferometric synthetic aperture radar (InSAR) can be used to measure wide-area surface deformation at the millimeter scale [31]. The insensitivity of SAR images to weather and the fact that the European Space Agency (ESA) has made the C-band Sentinel-1A data available for free since 2014 enables the periodic acquisition of SAR images [31–35].

The reservoir area of the Benzilan-Gangtuo ten-stage hydropower station on the UJSR was selected as the study area. The identification of active landslides was first realized step-by-step using comprehensive remote sensing technologies such as stacking-InSAR technology, optical satellite remote sensing images, and UAV images. Second, SBAS-InSAR technology was used to obtain the time-series deformation curves of active landslide samples to reveal the deformation types of active landslides. These research results are able to provide guidance for the monitoring and early warning of active landslides in the UJSR, serve the construction of major projects, and act as a reference for the study of active landslides in the basin.

## 2. Study Area

The study area lies on the southeastern rim of the QTP and the UJSR, extending from Penziban town in Deqin County, Yunnan Province to Gangtuo town in Gangda County, Tibet. It encompasses nine county-level administrative areas, comprising Shangri-La and Deqin in Yunnan Province; Derong, Batang, Baiyu, and Dege in Sichuan Province; and Mangkang, Gongjue, and Jiangda in Tibet. The main stream of the JSR has a river channel length of approximately 535 km, covering an area of approximately 18,294 km$^2$ (Figure 1a,b). The study area is situated in the transitional area between the first and second topographic steps in China, located in the northern part of the Hengduan Mountains. Elevations range from 1914 to 5823 m, with a gradual descent in the terrain from north to south. The hydropower plan for the UJSR states that the hydropower stations currently planned and under construction will adopt a "one reservoir and thirteen stages" layout. From downstream to upstream, the hydropower stations are Benzilan, Xulong, Changbo, Suwalong, Batang, Lawa, Yebatan, Bolo, Yanbi, Gangtuo, Guotong, Saila, and Xirong [5]. The study

area spans the ten hydropower station levels from Benzilan to Gangtuo (Figure 1c). Due to the complex topography, the climate in the area is varied and includes tropical monsoon, subtropical monsoon, and plateau monsoon climates. The study area experiences an annual rainfall of approximately 850 mm to 1500 mm (Figure 1b). The area's climate is characterized by distinct wet and dry seasons, with the rainy season occurring from June to September annually. The stratigraphy, shown in Figure 1d, spans from the Proterozoic (Pt) to the Quaternary (Q), with the main stratigraphic ages being Triassic (T), Permian (P), Carboniferous (C), and Devonian (D). Lithologies in the area predominantly comprise slate, shale, gneiss, sandstone, and limestone. Influenced by the uplift of the QTP, the area under investigation is tectonically complex (Figure 1e). It has developed numerous deep and broad fracture zones, such as the Jinsha-Honghe fracture zone, the Delai-Dingqu fracture zone, and the Zigasi-Yangla fracture zone [36]. Seismic events occur frequently. According to statistics on the website of the US Geological Survey (USGS), earthquakes in the study area are more concentrated near Deqin and Batang. Between 24 May 1929 and 1 February 2019, a total of 62 earthquakes of varying magnitudes occurred in the study area, 23 of which were magnitude 3 or above.

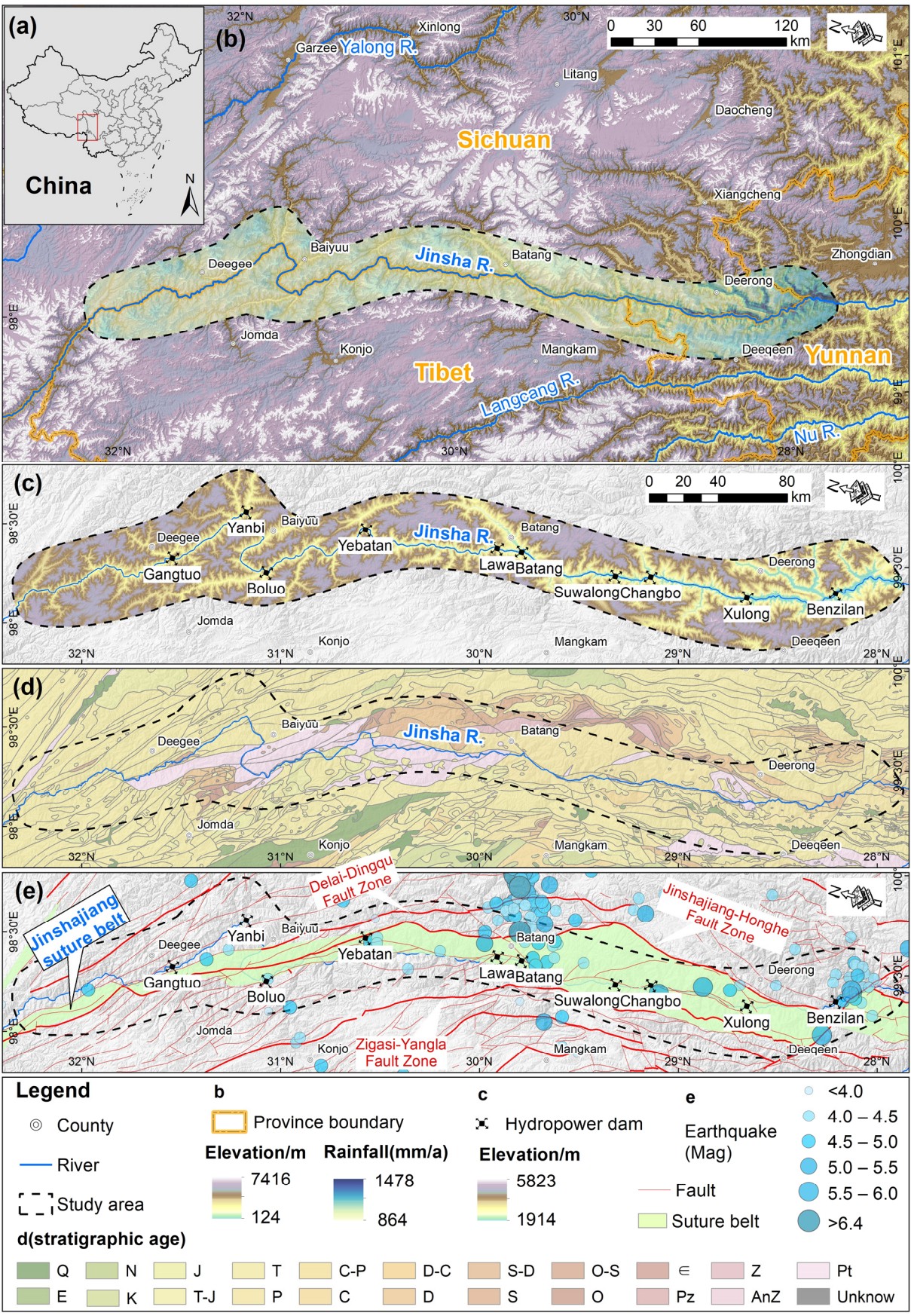

**Figure 1.** Map of the study area. (**a**) Location of the study area in China; (**b**) administrative divisions and rainfall map; (**c**) map of topography and hydropower stations; (**d**) stratigraphic map; (**e**) tectonic

and seismic map. The topographic elevation data are from STRM, 30 m DEM; the rainfall data are from National Tibetan Plateau/Third Pole Environment Data Center (Reprinted/adapted with permission from Ref. [37]. 2020, Shouzhang, P.); the stratigraphic dataset is from the National Geological Archives of China; and the historic earthquake data are from USGS.

## 3. Data

### 3.1. SAR Images and Visibility

#### 3.1.1. SAR Images

We collected radar images from Sentinel-1A, including path 99 ascending images, path 33 descending images, and path 106 descending images, taken over the study area (Figure 2). The SAR dataset's basic parameters and data volume are listed in Table 1. To reduce external errors, the precise orbit data corresponding to the time of each Sentinel-1A image view were downloaded for this research. These data were used for precision alignment and the elimination of systematic errors resulting from phase orbit errors. Meanwhile, the ALOS DEM data with a 12.5 m resolution for the study area were acquired to remove the terrain phase during InSAR processing, aid geocoding, and determine the overlay mask and shadow area.

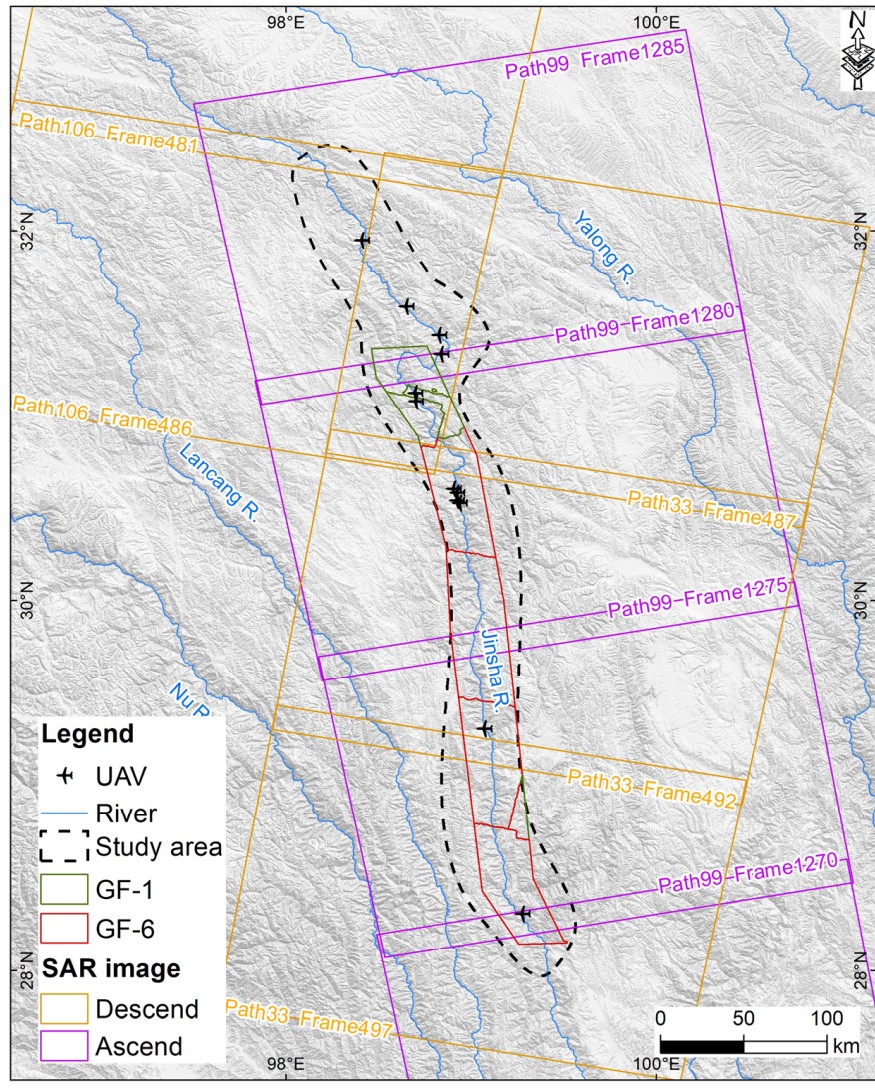

**Figure 2.** Map of multi-source remote sensing data coverage.

**Table 1.** Information on SAR data parameters in the study area.

| Orbital Direction | Number of Path | Number of Frame | Data Span | Volume of Images |
|---|---|---|---|---|
| Ascending | 99 | 1285/1280 | 12 January 2018–19 August 2022 | 248 |
| | | 1275/1270 | 12 January 2018–30 October 2022 | 266 |
| Descending | 33 | 497/492/487 | 7 January 2018–6 November 2022 | 429 |
| | 106 | 486/481 | 12 January 2018–11 November 2022 | 192 |

### 3.1.2. Visibility Evaluation

Alterations in terrain relief will impact the sequence in which the ground-reflected radar signals reach the imaging system, leading to geometric abnormalities such as perspective contraction, shadowing, and superimposed masking in the SAR image. The objective of visibility evaluation was to statistically determine the distributions of the data from the Sentinel-1A ascending and descending trajectories that covered the visible study area. The study area contained masked and shadowed areas, as shown in Figure 3a,b, respectively, alongside the distribution of visible and nonvisible regions in the joint monitoring of Sentinel-1A ascending and descending tracks (Figure 3c).

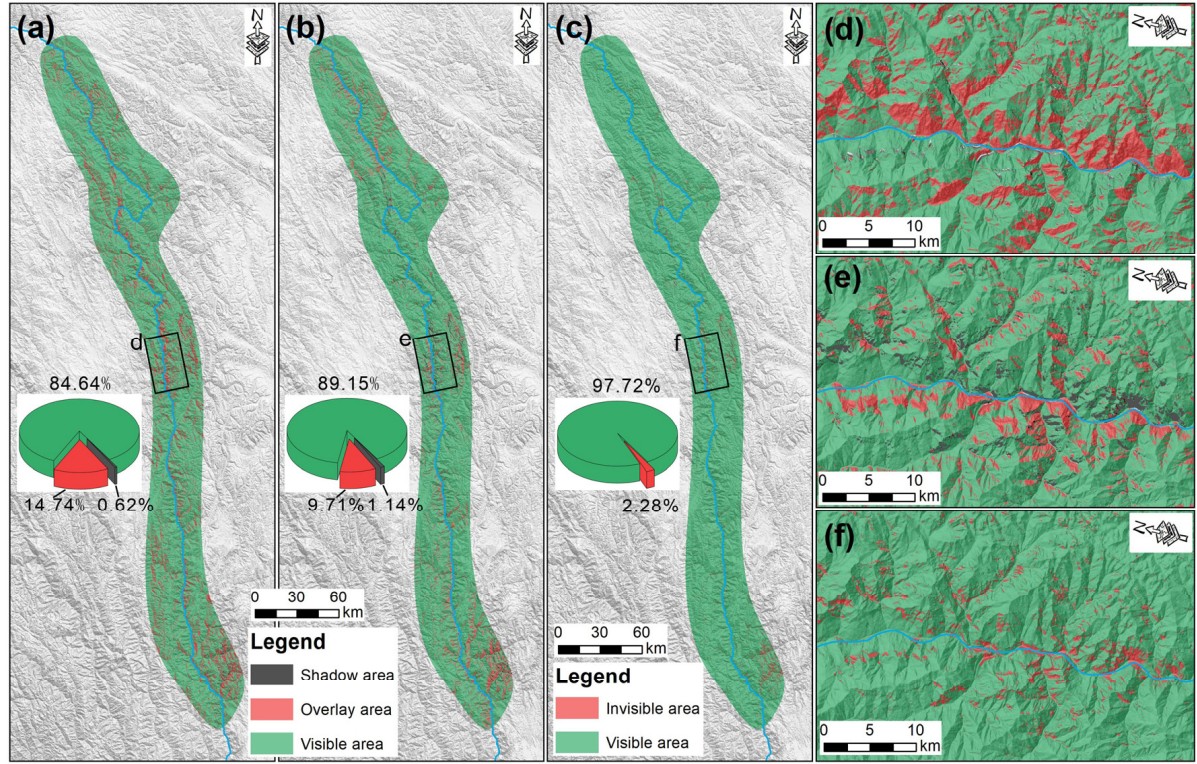

**Figure 3.** Results of visibility evaluation in the study area. (**a**) Ascending orbit data; (**b**) descending orbit data; (**c**) combined ascending and descending orbit data; (**d**–**f**) local zoomed-in view.

The results of the SAR visibility analysis indicate that the study area has visible areas comprising 84.64% of the Sentinel-1A ascending data and 89.15% of the descending data. The area covered by the superposition mask is more concentrated along the two sides of the main stream of the JSR, with superposition mask ratios of 14.74% and 9.71%, respectively. Notably, based on a combination of Sentinel-1A ascending and descending orbit data for detecting the study area, the visible area accounts for 97.72%, while the area of superimposed masks and shaded regions accounts for only 2.28%. This approach can effectively compensate for the limitations of single-orbit detection and significantly increase visibility within the study area, which meets the detection requirements more effectively.

*3.2. Optical Satellite Images*

With attention to the coverage and precision of the optical satellite images, GF-1 and GF-6 optical satellite image data with a 2 m resolution were gathered, covering a portion of the study area. Figure 2 displays the collected data. The identification of spectral characteristics of active landslides was challenging using a single image. To overcome this obstacle, we aimed to select images from various periods whenever possible. Limited to the availability and cloud coverage ratio of archived images (<5%), the acquisition dates for GF-1 images were 1 February 2019 and 13 February 2019, and those for GF-6 images were 17 August 2019, 15 October 2019, and 1 January 2020. Historical images from Sky Map (91 weitu v19.3.4) and Google Earth Pro (v7.3.6.9345) were collected for areas not within the coverage area of the GF-1 and GF-6 images.

*3.3. UAV Images*

UAV data were primarily employed to acquire the topography and geomorphology of the landslides, the indications of deformation, and the structural attributes of the slopes. On the one hand, the study utilized UAV-inclined photogrammetry and 3D modelling techniques to procure 3D models of eight active landslide hazards spanning a total area of 19.3 km$^2$. The UAV image was taken at a relative height of 850 m with a resolution of approximately 0.2 m. On the other hand, light detection and ranging (LiDAR) techniques provided data from airborne LiDAR point clouds of four active landslide hazards spanning a total area of 7.1 km$^2$. The digital orthophoto map (DOM) and digital elevation model (DEM) data were acquired simultaneously and have a 0.1 m resolution.

**4. Methods**

The workflow in this study is as follows: initially, the deformation area was identified through stacking-InSAR. Then, we eliminated non-landslide deformation and identified active landslide boundaries by analyzing morphological features and macro deformation based on high-resolution optical satellite images. Subsequently, field investigations were conducted using UAVs, airborne LiDAR, and traditional methods to confirm landslides. Finally, SBAS-InSAR technology was used to obtain the time-series deformation curves of the strong deformation zones of the landslides, and the active landslides were classified based on the curve morphology (Figure 4).

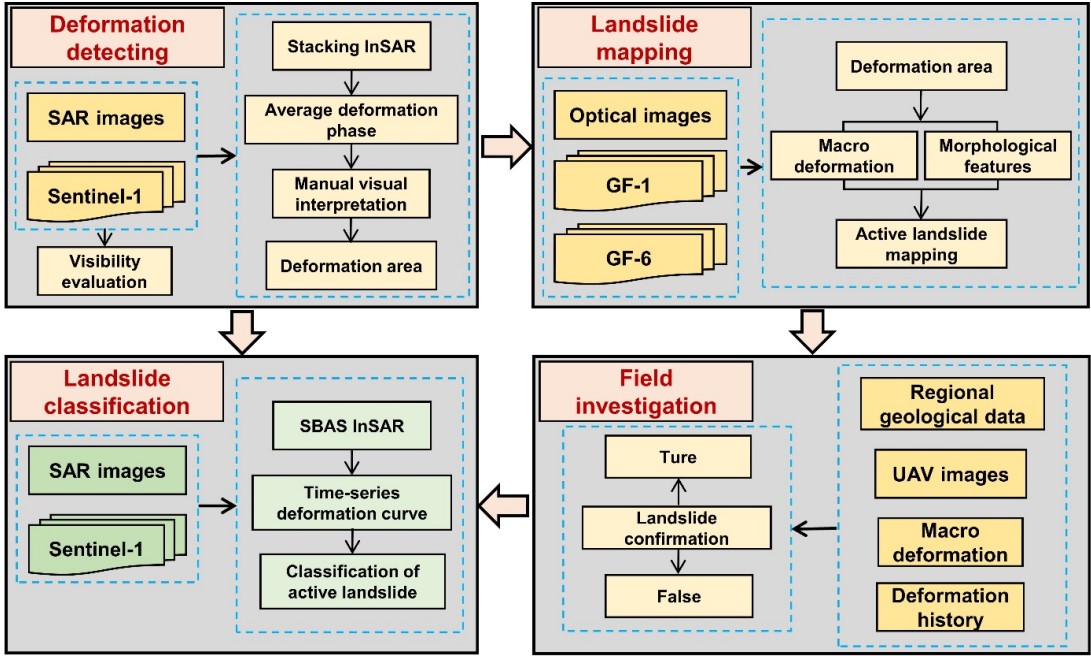

**Figure 4.** Active landslide identification flowchart.

*4.1. Identification of Active Landslides*

Active landslides are slopes that display macroscopic or microscopic deformation. Typically, macroscopic signs can be identified through conventional field investigations [22]. In China, more than 65% of the land is in mountainous regions, and many active landslides originate at high elevations, increasing the challenge of early detection. Currently, remote sensing technology is extensively utilized in analyzing the spatial distribution of landslides. It integrates data from multiple sources to obtain more accurate landslide information [38].

4.1.1. Deformation Detection

InSAR technology can continuously monitor ground deformation across a large area and is effective in identifying landslides situated in remote or sparsely populated regions [22]. Conventional differential InSAR (DInSAR) is prone to atmospheric delays and incoherence, resulting in substantial inaccuracies in deformation measurements. However, stacking-InSAR is a weighted averaging of de-entangled phases obtained via DInSAR. It can effectively ameliorate the impacts of random orbital errors, topographic errors, and atmospheric phase delay errors. It can better capture the range of deformation and morphological characteristics of deformation in space and is capable of detecting landslides with smaller magnitudes of deformation. Hence, stacking-InSAR is appropriate for monitoring surface deformation in mountainous terrain [39].

Figure 5 displays a representative stacked phase map of the research area produced by stacking-InSAR technology. In the stacked phase map, numerous regions exhibit dense interference fringes and have significant variations in color. These are interpreted as deformation anomalies (yellow circled area). The white area in the image represents an invisible area.

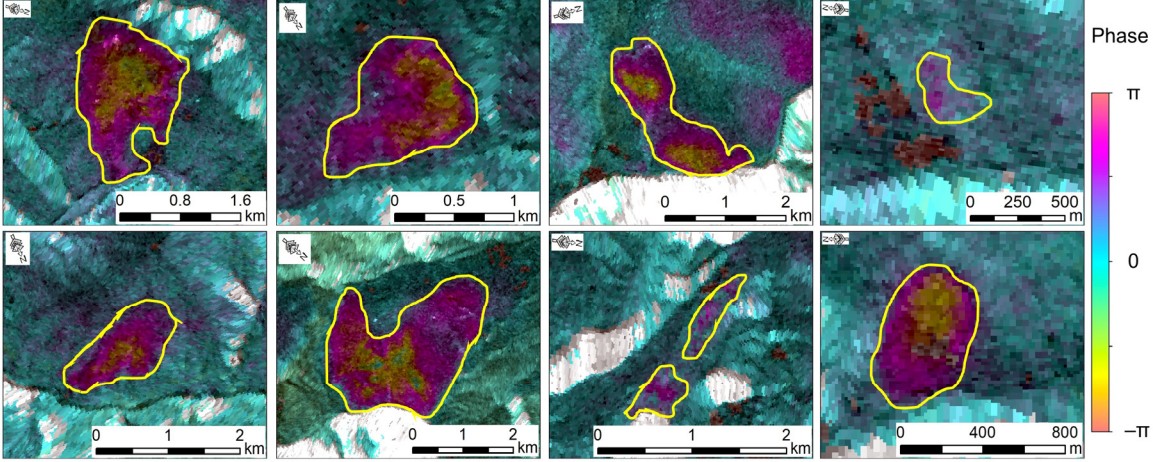

**Figure 5.** Stacked phase map created using stacking-InSAR technology (the yellow circled area is the deformation area).

4.1.2. Landslide Mapping

1. Non-landslide Deformations

It is assumed that variables such as vegetation cover, topographical effects, and inaccuracies in data processing are fully eliminated. In addition to landslides, ground surface deformation frequently results from snow and ice melting, anthropogenic activities, etc. Consequently, the deformation region identified by stacking-InSAR technology should not be immediately regarded as a landslide. We excluded three types of non-landslide deformations observed in the optical satellite images. The first type was the deformation resulting from the melting of snow and ice, which was evident as a clustered strip area in high-elevation regions (generally greater than 4500 m) with steep gradients (Figure 6a,b). The second type of deformation involved dispersed small-scale changes. This included the

source area of a debris flow (Figure 6c,d), seasonal changes on the surface due to variations in the water content of the loose sediment, and alterations to the ice layer on the lake surface, among others. The third type of deformation was caused by human engineering activities, primarily targeting construction sites with low-relief topography and gentle slopes (Figure 6e,f).

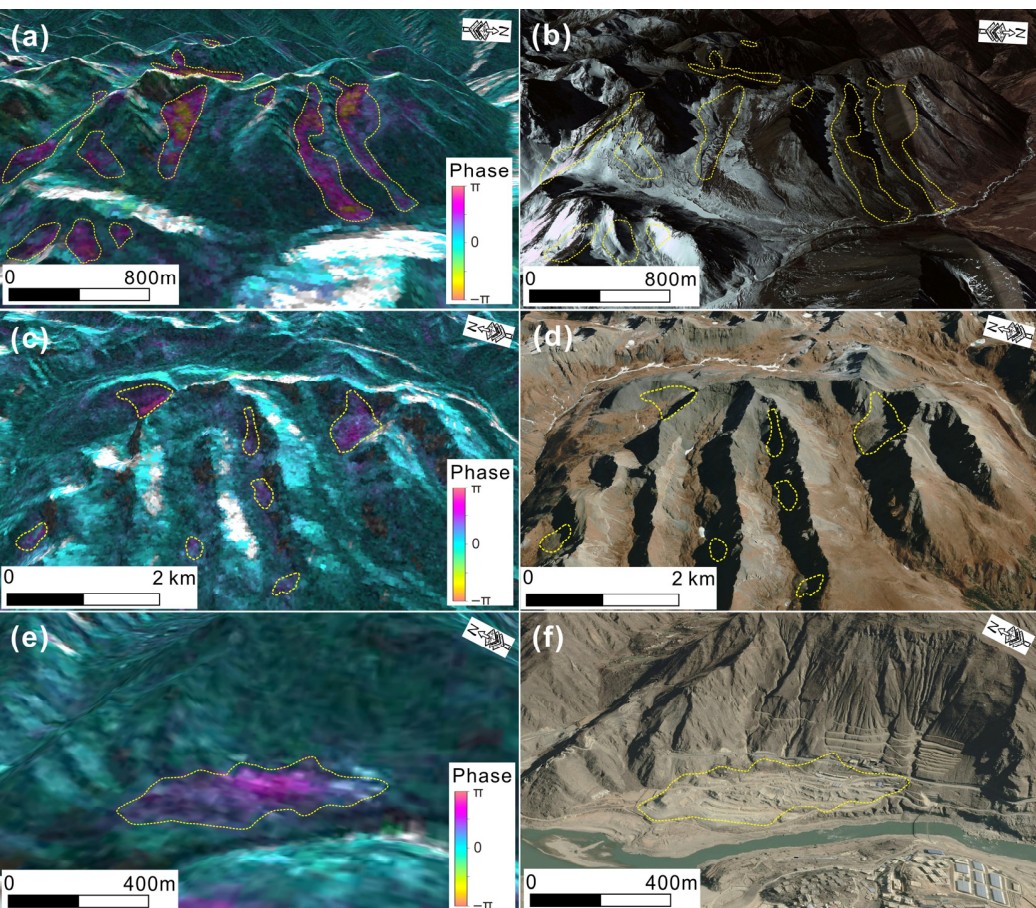

**Figure 6.** Examples of InSAR deformation rejected in the study area. (**a**,**b**) The deformation caused by melting snow and ice. (**c**,**d**) The dispersed small deformation located at the source area of the debris flow. (**e**,**f**) The deformation triggered by human engineering activities.

2.  Landslide Characteristics in Optical Satellite Images

After removing non-landslide deformation, the focus of this study was on the deformation caused by active landslides. Nonetheless, it is important to note that the deformation boundary could not be directly applied as the landslide boundary since the InSAR technique may detect only a section of the landslide, which does not represent its complete shape. For instance, we used the stacking-InSAR technique to acquire a stacking phase map of the landslide in Guxue village. This highlighted two elliptical and strong deformation zones (Figure 7a). We also discovered that the slope on which Guxue village is situated appeared to be a complete ancient landslide stack upon examination of the optical satellite image (Figure 7b). Ultimately, we delineated the boundary of the Guxue landslide and identified the strong deformation zone detected through InSAR as an ancient landslide reactivation. Therefore, it is necessary to define the boundaries of landslides using optical satellite imagery.

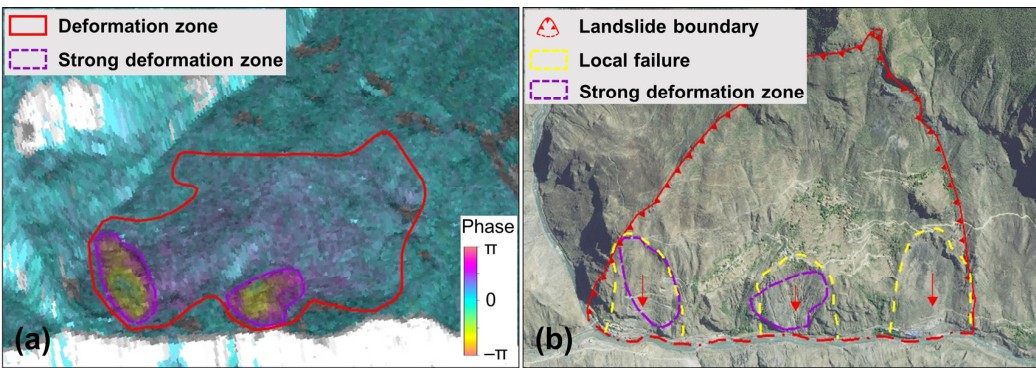

**Figure 7.** Guxue landslide deformation zones and optical image.

According to the history of landslide activity, active landslides in the study region can be broadly classified into two types: reactivated ancient landslides and landslides in gestation. Ancient landslides generally refer to those with a lengthy formation period, and they are the result of a lengthy, complex evolutionary process of slopes and may reactivate due to human activities, heavy rainfall, and earthquakes [40]. Due to their age, ancient landslides generally do not differ greatly from their surroundings in color and tone. However, they have distinctive features in terms of microgeomorphology, such as landslide wall and body. The landslide wall is distributed at the rear and sides of the landslides, which are usually steep and high, and typically appear as shadows on optical satellite images (Figure 8a). The middle portion is usually broad, with a gentle topography and high vegetation cover, and is frequently utilized as cultivated land. Settlements and temples are also distributed in this zone (Figure 8b). The landslide body comprises the primary segment formed by the accumulation of landslide material. The topography of the leading edge becomes steeper and more pronounced than the surrounding terrain. This squeezes the river channel as it joins the river and forms what is known as a "landslide tongue" (Figure 8c). The morphological features of a landslide in gestation are not always apparent and typically lack a distinct chair-like shape. Signs indicating its presence are primarily manifested through deformation features such as cracks (tension and shear), avalanches, and slide damage. Differences in color, shadow, and texture when compared to the surrounding environment are also noticeable, with predominantly grey-brown and greyish-white tones and rough textures. The tensile cracks primarily appear in the central and rear regions of the landslide. On optical satellite images, these cracks exhibit mostly curved and jagged shapes on both flanks. Shear cracks are predominantly situated in the shear outlets and sidewalls at the leading edge of the landslide and exhibit an irregularly curved and jagged appearance in the image (Figure 8e). The development of a multistage landslide bench is attributed to the further progression of tensile cracks, primarily distributed in the central and rear sections of the landslide and characterized as bench-shaped (Figure 8f). The outcomes of boundary delineation of typical active landslides in the study region are presented in Figure 9.

4.1.3. Field Investigation

After completing the mapping of landslides in the study area, field investigations typically proceed using UAV aerial photography and airborne LiDAR in conjunction with regional geologic data. The primary investigations include the following. (1) Geological environmental conditions, such as slope structure and stratigraphic lithology. (2) Landslide boundaries and macrodeformation indicators. (3) The impact of external factors, including rainfall, earthquakes, floods, and human engineering activities, on landslides. (4) This section evaluates the effectiveness of current monitoring equipment, treatment facilities, and other measures used for the prevention and control of landslides. (5) A preliminary assessment is conducted to determine the stability of active landslides. Figure 10 illus-

trates the field investigation drone and camera photographs depicting a typical landslide, specifically the Aluogong landslide.

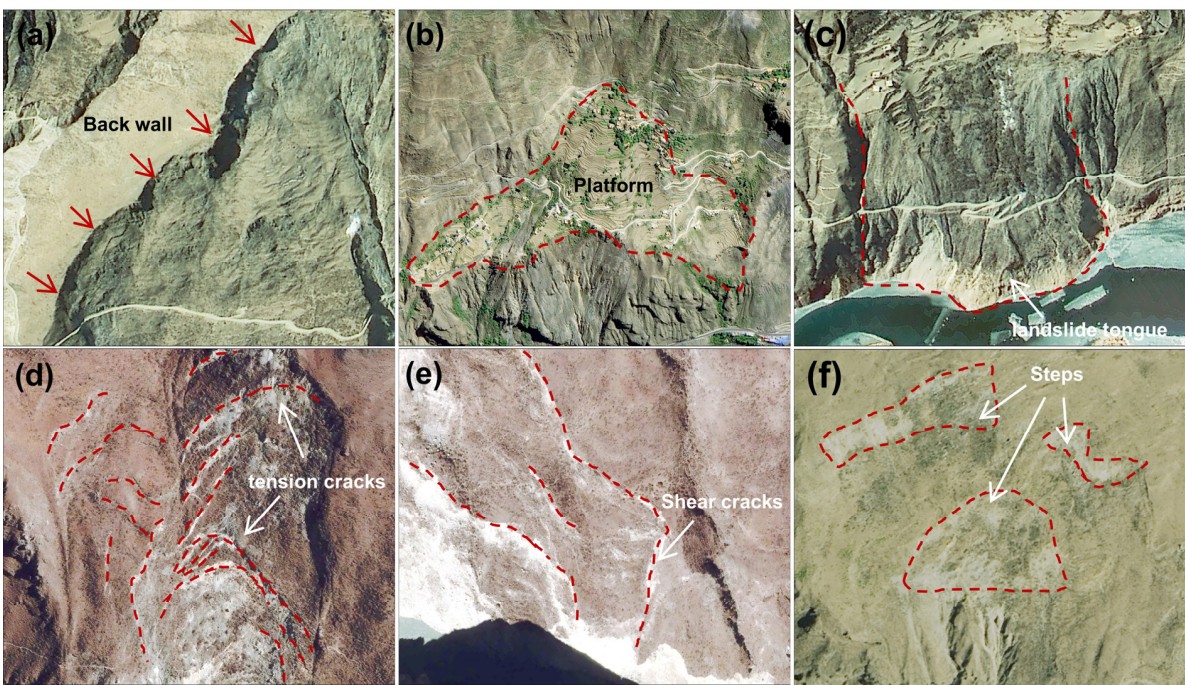

**Figure 8.** Landslide optical image features. (**a**) Back wall; (**b**) local platform; (**c**) landslide tongue; (**d**,**e**) tension cracks and shear cracks; (**f**) steps.

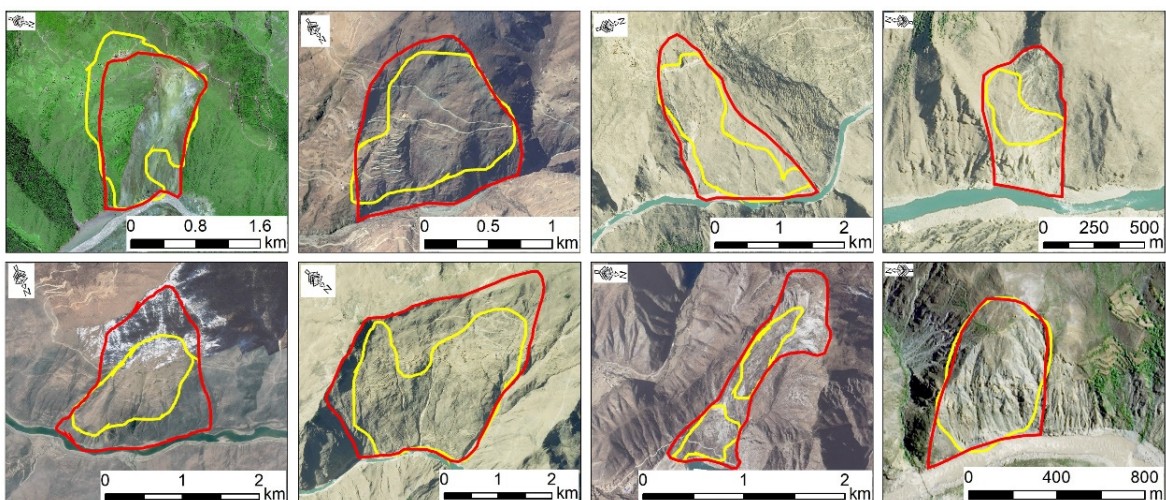

**Figure 9.** Typical landslide boundaries in the study area (the yellow circled area is the deformation area and the red circled area is the landslide boundary).

## 4.2. Time-Series Deformation Analysis of Active Landslides

Numerous studies have demonstrated that SBAS-InSAR technology is more applicable to valley areas than other time-series radar interferometry technologies [38,41,42]. The basic process of SBAS-InSAR is as follows. Initially, SAR images in single look complex (SLC) format are used to establish an interferometric pair network by applying a specific spatiotemporal baseline threshold. Then, differential interference processing is performed on each interference image pair to obtain the correct unwrapping phase. Next, the least square method or singular value decomposition (SVD) is used to calculate the deformation phase and residual phase of the model. Finally, the atmospheric delay phases and nonlinear

deformation are separated through filtering techniques [31]. Figure 11 displays the average annual deformation rate of a representative region detected by SBAS-InSAR technology in combination with ascending and descending Sentinel-1A radar images. This region has 17 landslides, and L106 is one of the largest.

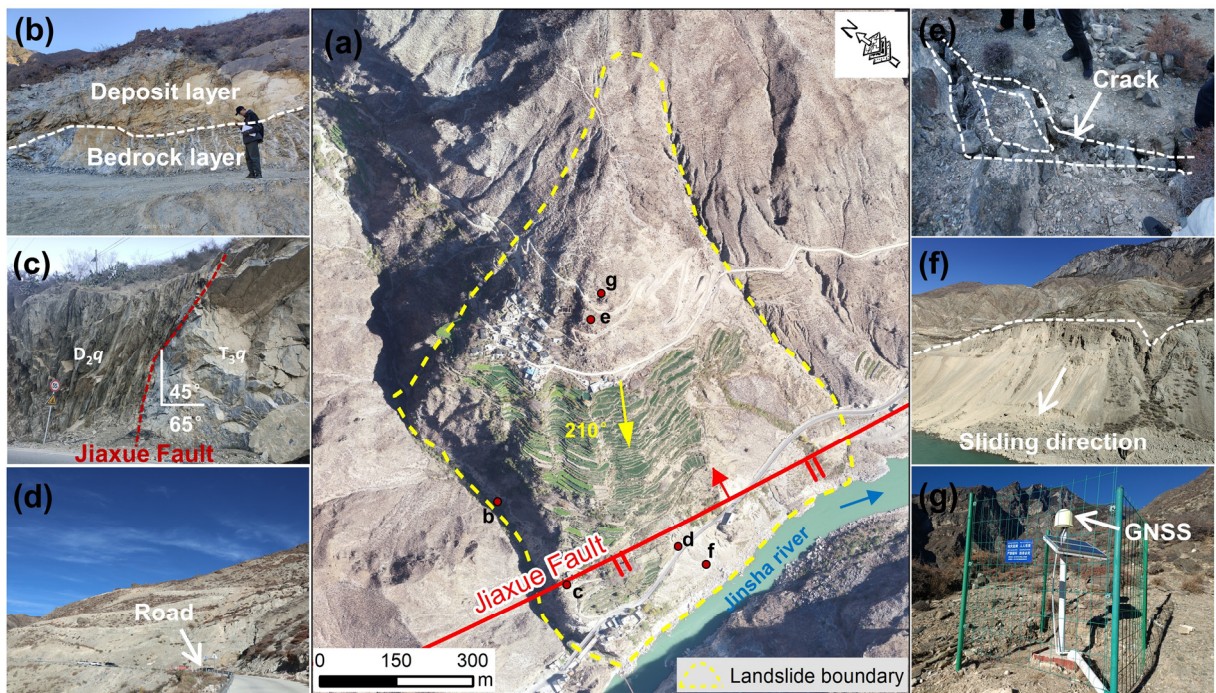

**Figure 10.** Field investigation of typical landslide. (**a**) UAV image; (**b**) bedrock cover interface; (**c**) Jiaxue fault; (**d**) slope cutting; (**e**) crack; (**f**) local sliding failure; (**g**) monitoring equipment.

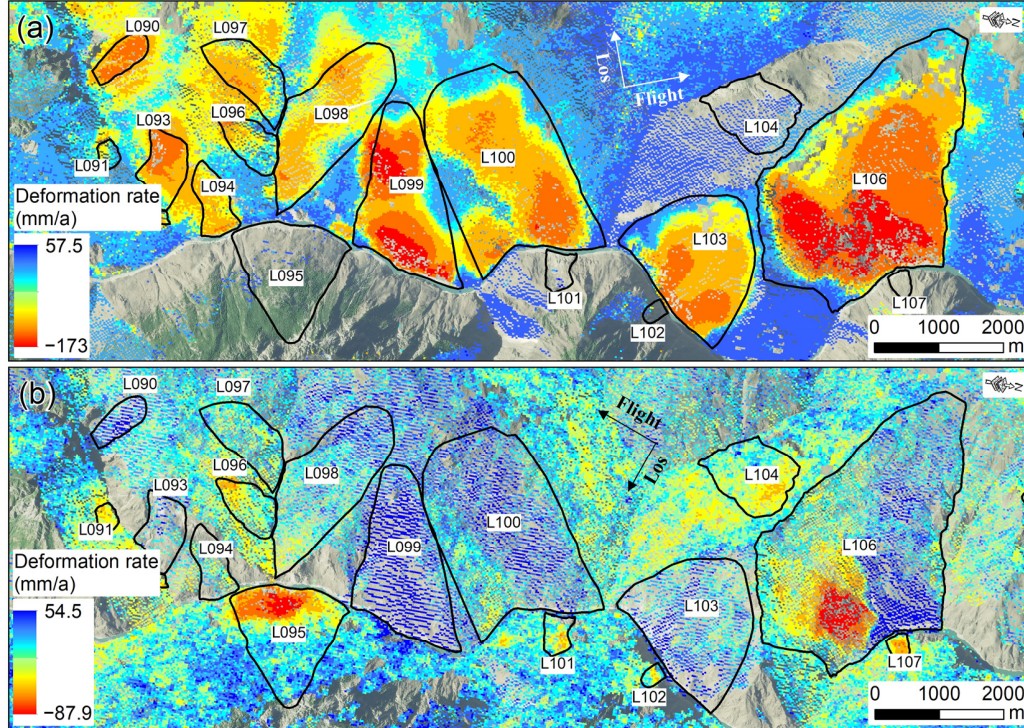

**Figure 11.** Average annual deformation rate for a representative region. (**a**) Deformation rate of ascending images; (**b**) deformation rate of descending images.

## 5. Results

### 5.1. Identification of Active Landslides

A total of 246 active landslides were identified in the study area, and the spatial distribution is shown in Figure 12a. The relationship between active landslides and reservoir areas is as follows: 5 in front of the Benzilan Dam, 14 in the Benzilan Reservoir area, 44 in the Xulong Reservoir area, 7 in the Changbo Reservoir area, 13 in the Suwalong Reservoir area, 1 in the Batang Reservoir area, 29 in the Lawa Reservoir area, 14 in the Yebatan Reservoir area, 57 in the Boluo Reservoir area, 35 in the Yanbi Reservoir area, and 27 in the Gangtuo Reservoir area. The results of Gaussian kernel density analysis of landslides are shown in Figure 12b. They indicate the presence of three zones with concentrated active landslide occurrences (zones I, II, and III) in the study area. Table 2 provides an overview of the three zones' basic features. These zones experienced 207 landslides, accounting for 84.1% of the total landslides in the area. The average landslide density is about $32.9/10^3$ km². Zone I is situated in the Xulong–Suwalong region, covering an area of approximately $1.5 \times 10^3$ km². This zone had 51 landslides in total, with an average density of approximately 34 landslides/$10^3$ km². This accounts for 24.6% of the landslides in the three areas and 20.7% of the total number of landslides in the study area. The Wangdalong landslide is a typical example of landslides that have occurred in Zone I. Zone II is situated in the Yebatan region and covers an area of approximately $1.1 \times 10^3$ km². The number of landslides recorded in this zone is 29, with an average density of approximately 26.4 landslides per $10^3$ km², constituting 14.0% of the total landslide count in the three areas and 11.8% of the overall landslide count in the study area. Typical landslides found in Zone II include the Xiongba landslide. Zone III is situated within the Yanbi–Gangtuo section, covering approximately $3.6 \times 10^3$ km². This zone had 127 landslides in total, with an average density of approximately 32.9 landslides/$10^3$ km². This represents 61.4% of all landslides in the three areas and 51.6% of landslides in the study area. Additionally, typical landslides, such as the Baige landslide and Woda landslide, were observed.

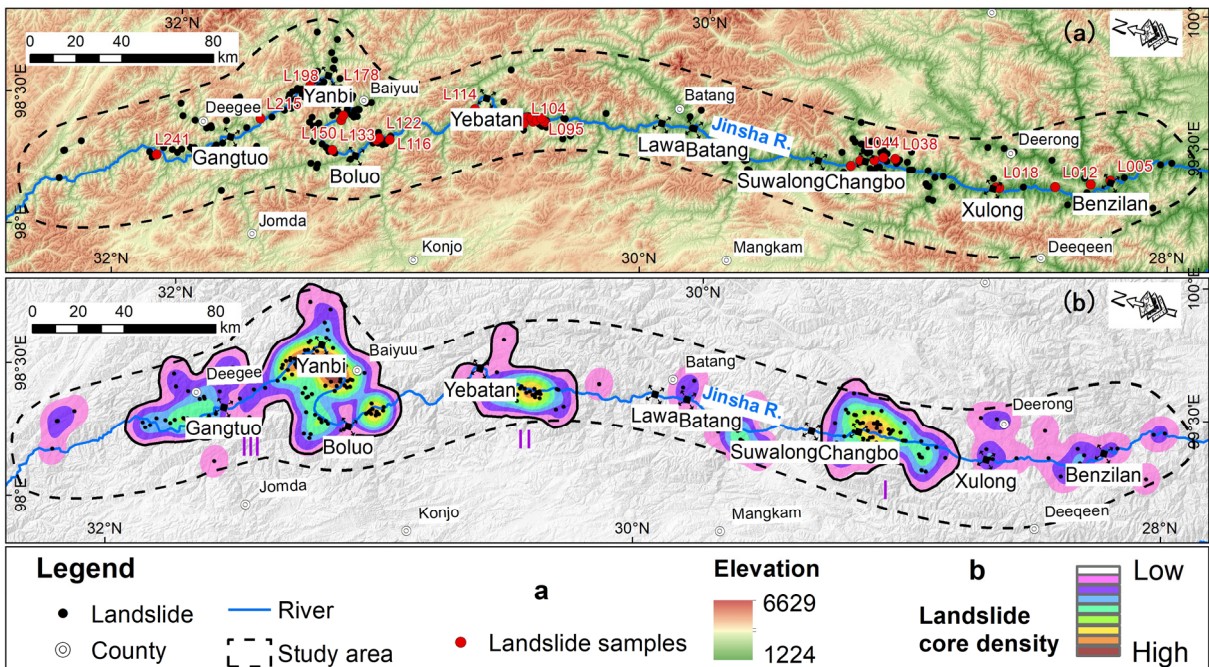

**Figure 12.** Map of active landslide identification. (**a**) Basic spatial distribution characteristics; (**b**) Non-uniformed spatial distribution and landslide concentration zones.

**Table 2.** Statistical table of concentration zones of active landslides.

| Zone | I | II | III | Total |
|---|---|---|---|---|
| Area ($\times 10^3$ km$^2$) | 1.5 | 1.1 | 3.6 | 6.3 |
| Number of landslides | 51 | 29 | 127 | 207 |
| Average density ($/10^3$ km$^2$) | 34 | 26.4 | 35.3 | 32.9 |
| Percentage of landslides in centralized zones | 24.6% | 14.0% | 61.4% | 100% |
| Percentage of total landslides | 20.7% | 11.8% | 51.6% | 84.1% |
| Typical landslides | Wangdalong landslide | Xiongba landslide | Baige landslide, Woda landslide | |

### 5.2. Deformation Types of Active Landslides

#### 5.2.1. Linear Type

L064 is situated on the left bank of the JSR, 2 km upstream of the Changbo hydropower station. L148 is located on the right bank of the JSR, 30 km upstream of Boluo Hydropower Plant. Deformation values of points P1, P2, P3, and P4 were obtained within the strong deformation zones of landslides L064 and L148. Figure 13 shows the curves, indicating a near-linear progression with an essentially identical rate of deformation, while the cumulative deformation gradually increases. Yearly average deformation rates of 22.3 mm/a and 21.2 mm/a were observed for points P1 and P2. Similarly, it was found that the average annual deformation rates for P3 and P4 were 89.4 mm per year and 87.6 mm per year, respectively.

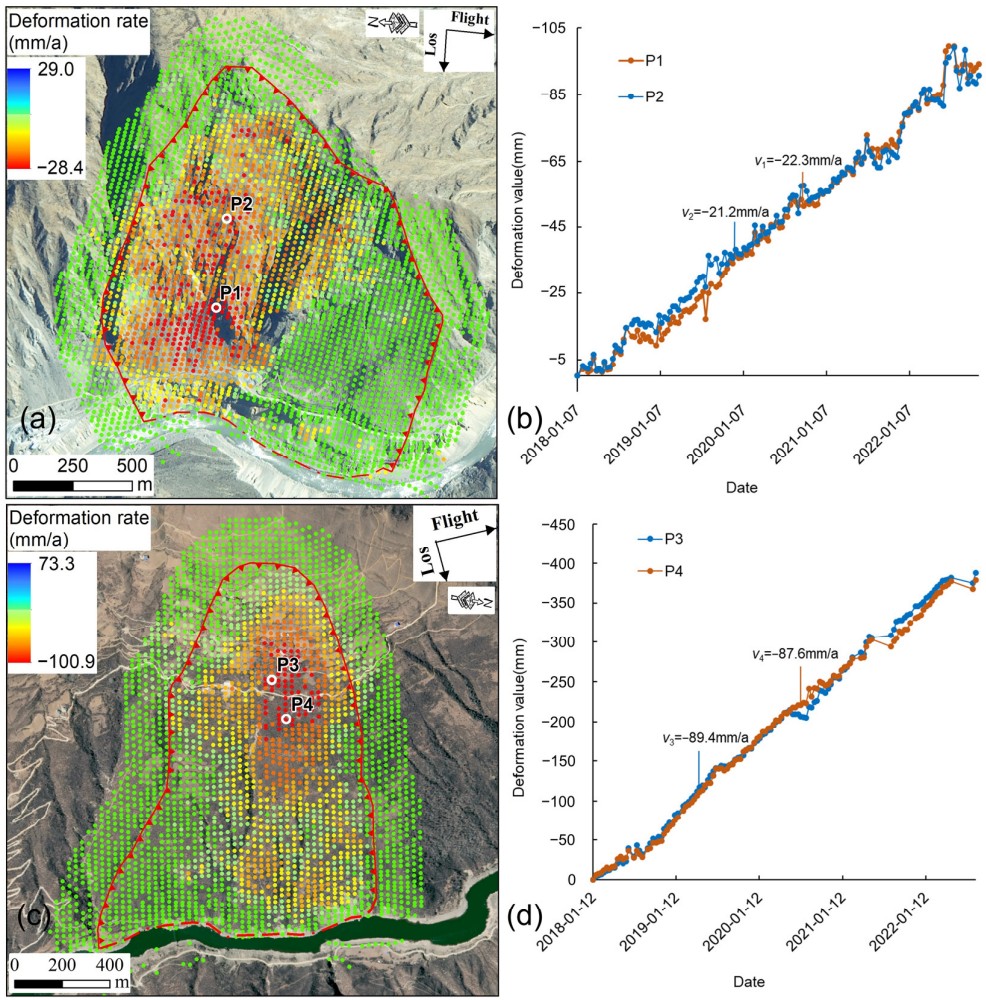

**Figure 13.** (**a**) Descending SBAS-InSAR map of L064; (**b**) time-series deformation curves at points P1 and P2 (horizontal coordinate is the date of the data acquisition; vertical coordinate is the inverse of

the cumulative deformational variables, the same as below); (**c**) ascending SBAS-InSAR map of L148; (**d**) time-series deformation curves at points P3 and P4.

### 5.2.2. Upward Concave Type

L018 is situated 1 km downstream from Xulong Hydropower Station on the right bank of JSR in the Benzilan reservoir area. L106, known as the Xiongba landslide, is positioned 20 km downstream from Yebatan hydropower station and on the left bank of the JSR. It is evident that these two landslides are experiencing accelerated deformation, characterized by the exponential increase in deformation value, based on the time-series deformation curves for points P5, P6, P7, and P8 situated in the strongly deformed zones of landslides L018 and L106 (Figure 13). Taking the case of the characteristic point P7 on L106, from January to October 2018, a gradual deformation was observed, culminating in a total deformation of about 27 mm at a rate of roughly 36.0 mm/a. Between October 2018 and March 2019, the deformation at P7 demonstrated an increasing trend affected by the Baige landslide incident, resulting in a cumulative deformation of 74 mm and a deformation rate of about 128.4 mm/a. Since March 2019, P7 has experienced uniform deformation. By the end of the monitoring period, the cumulative deformation had increased to approximately 668 mm, with an average deformation rate of around 170.5 mm per annum. We collected GNSS data from December 2021 to August 2022 and calculated using them for the radar line of sight. Figure 14e illustrates a strong correlation between the GNSS data and Point P7, with a correlation coefficient (CC) of 0.933. Presently, the curve displays an upward trend without any indication of convergence.

### 5.2.3. Downward Concave Type

L116, known as the Baige landslide, lies 18 km downstream of the Polo hydropower station, whereas L122 is located 4 km upstream of L116 on the right bank of the JSR within the Yebatan reservoir area. Figure 15 depicts the time-series deformation curves of points P9, P10, P11, and P12 in the strong deformation zones of landslides L116 and L122. The curves show a downward concave shape, gradually decreasing in steepness after a sharp increase. Taking the example of the characteristic point P9 on L116, it can be observed that the deformation of this point was slow between January and October 2018, with a total deformation of merely 8 mm and an average deformation rate of approximately 10.7 mm/a. However, from October 2018 to October 2019, the deformation of point P9 increased steeply, with a total deformation of 194 mm and an average deformation rate of approximately 190.7 mm/a. Subsequently, from October 2019 to October 2020, there was a cumulative deformation of only 8 mm at the monitoring point. From October 2020 to October 2021, the monitoring point experienced a cumulative deformation of approximately 163 mm, with an average deformation rate of approximately 160.2 mm/a. During the same period, the monitoring point showed a deformation of 93 mm, with an average deformation rate of approximately 94.7 mm/a. As of October 2021 (August 2022), the cumulative deformation measures 58 mm, with an average deformation rate. It is clear that the initial slide of L116 took place on 11 October 2018, which explains the sharp increase in the deformation curve at point P9. With continued ground stress adjustments, the slope is now gradually stabilizing.

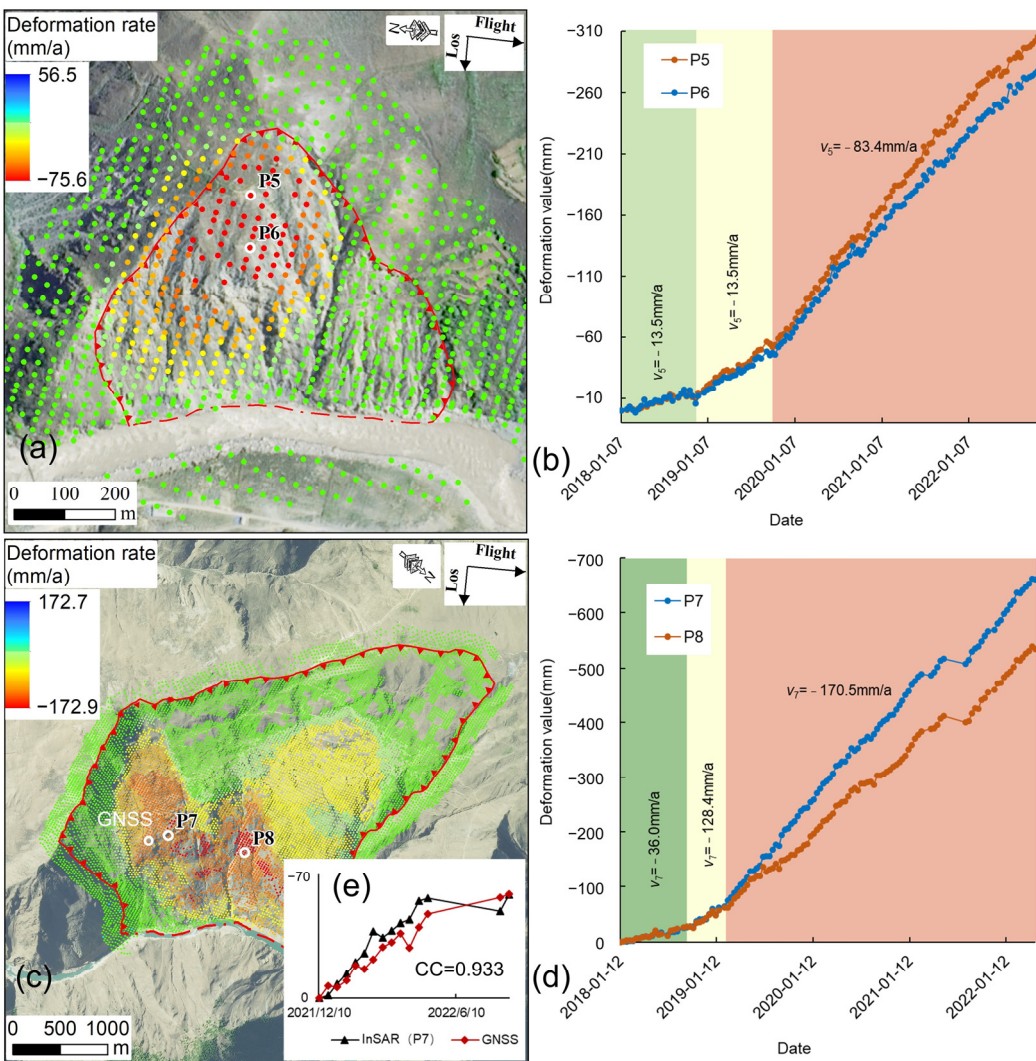

**Figure 14.** (**a**) Descending SBAS-InSAR map of L018; (**b**) time-series deformation curves at points P5 and P6 (the background color gradually transitions from green to red, indicating an increase in deformation rate; below is the same); (**c**) ascending SBAS-InSAR map of L106; (**d**) time-series deformation curves at points P7 and P8.

### 5.2.4. Stepped Type

L093 is located on the right bank of the JSR and belongs to the Lawa reservoir area. L197, known as the Woda landslide, is situated on the same bank of the JSR, around 9 km upstream of the Yanbi hydropower station. Deformation curves were acquired from points P13, P14, P15, and P16 within the strongly deformed zones of both L093 and L197. These curves exhibit a step shape and are divided into multiple deformation stages according to the difference in deformation rate (Figure 16). The highest average deformation rate observed for P13 was 108.7 mm/a, whereas the lowest was 9.3 mm/a. P15 showed a peak average deformation rate of 115.5 mm/a and a minimum of 0.05 mm/a.

### 5.2.5. Classification of Active Landslides

According to the time-series deformation curve, landslides can be categorized into four types: linear type, upward concave type, downward concave type, and stepped type (Figure 16). Linear-type landslides have a consistent deformation speed and a cumulative deformation curve that rises slowly with a stable growth trend. The slope may be in the initial deformation or isotropic deformation stage (Figure 17a). The rate of upward concave-type landslide deformation exhibits exponential growth, with a cumulative de-

formation curve forming a downward concave arc. The slope is in a stage of accelerated deformation, which generally poses a higher risk of landslide instability (Figure 17b). In contrast, downward concave-type landslide deformation rates display rapid growth that gradually slows down and ultimately stabilizes. The time-series deformation curve forms a downward convex arc (Figure 17c). The deformation curve for a time-series of stepped-type landslides resembles a ladder, with clear instances of acceleration followed by stabilization. The stepped-type shape of the curve may be attributed to the existence of a locking section of the sliding surface or the influence of external factors such as heavy rainfall, fluctuations in water levels, or cyclical effects (Figure 17d).

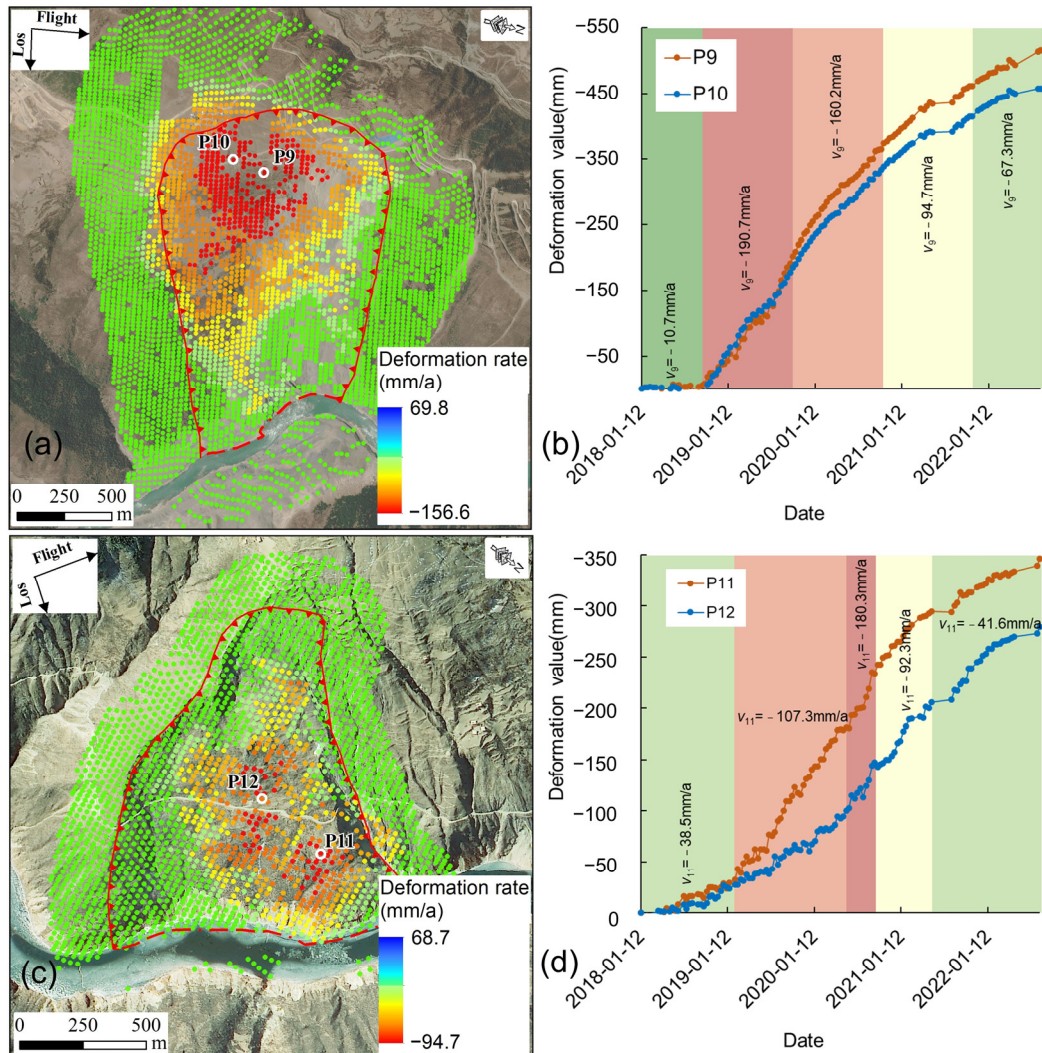

**Figure 15.** (**a**) Ascending SBAS-InSAR map of L116; (**b**) time-series deformation curves at points P9 and P10; (**c**) ascending orbit SBAS-InSAR map of L122; (**d**) time-series deformation curves at points P11 and P12.

Thirty-one landslides were selected as study samples in the examined area (Figure 12a). There were six linear-type landslides, three upward concave-type landslides, 10 downward concave-type landslides, and 12 stepped-type landslides (Table 3).

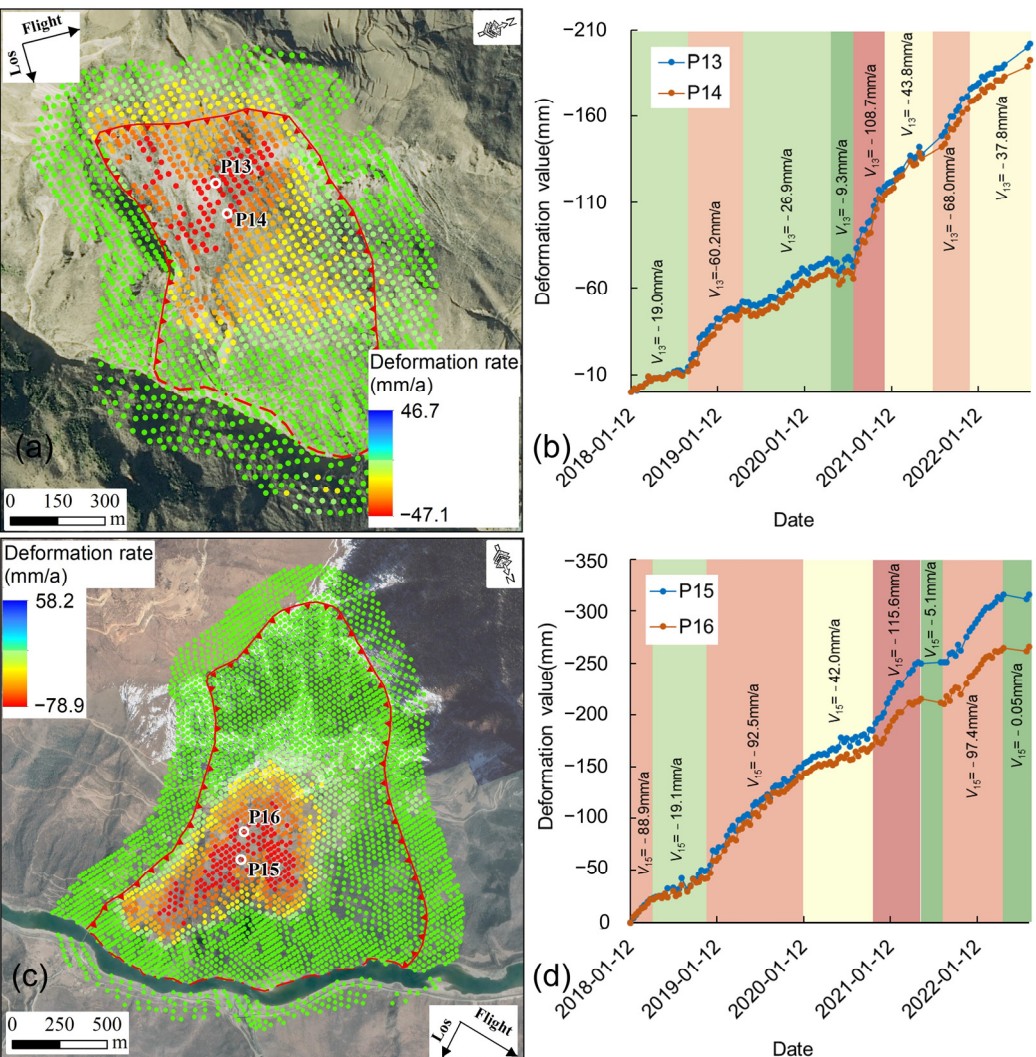

**Figure 16.** (**a**) Ascending track SBAS-InSAR map of L093; (**b**) time-series deformation curves at points P13 and P14; (**c**) ascending track SBAS-InSAR map of L197; (**d**) time-series deformation curves at points P15 and P16.

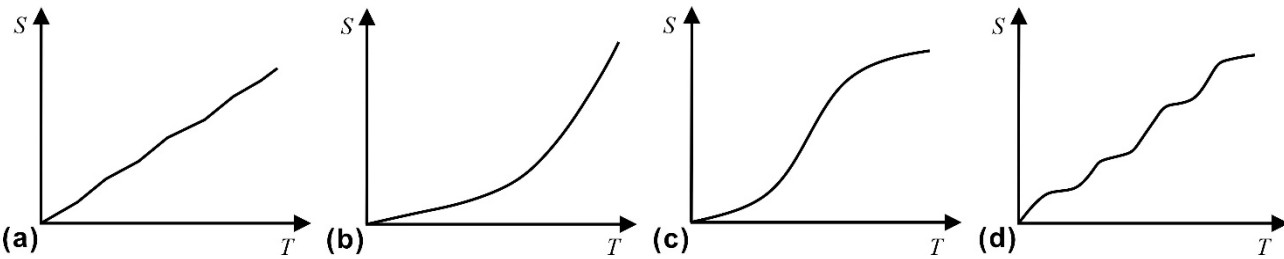

**Figure 17.** Classification of active landslide by time-series curves. (**a**) Linear-type landslide; (**b**) upward concave-type landslide; (**c**) downward concave-type landslide; (**d**) stepped-type landslide.

**Table 3.** Classification of active landslides.

| Type | Landslides |
|---|---|
| Linear type | L005, L017, L064, L068, L101, L148 |
| Upward concave type | L018, L052, L106(Xiongba landslide) |
| Downward concave type | L012, L038, L044, L094, L095, L099, L100, L114, L116(Baige landslide), L122 |
| Stepped type | L006, L093, L103, L104, L124, L133, L150, L178, L197(Woda landslide), L198, L215, L241 |

## 6. Discussion

Integrated remote sensing technology has proven effective in the UJSR. First, the combination of ascending and descending SAR images for deformation detection significantly increased the visible area in the study area. Second, the use of high-resolution optical satellite images effectively removed non-landslide deformation and delineated the landslide boundary. However, we gathered only C-band Sentinel-1A data. If we aimed to further decrease the rate of misjudgment and omission of active landslides, we could utilize multisource SAR data for deformation detection.

The SBAS-InSAR technique obtains the projection of the true landslide deformation in the line-of-sight (LOS) direction of the radar satellite, which is not equivalent to real landslide deformation and solely serves to determine its deformation trend. If InSAR is utilized to acquire actual landslide deformation, multiple-orbital radar satellite data can be employed for collaborative detection, resolving three-dimensional landslide deformation information.

The current classification of landslides only summarizes the trends of the surface deformation, without including geological background information such as landform, lithology, structure, geotechnical characteristics, landslide morphology, and deformation mechanism. Future research will combine detailed field investigations, indoor model testing, and numerical simulations to thoroughly study the factors that control landslide displacement [43,44].

According to C. Juez et al., sediments supplied over short time-scales tend to be large sediment contributors relative to the long-term time-scale mean [7]. Landslides deliver a large amount of sediment to rivers in the short-term, posing a challenge to reservoir management. For instance, the first and second Baige landslides alone delivered 23 million $m^3$ and 3.5 million $m^3$ of sediment, respectively, into the JSR [10]. This further highlights the significance of identifying the spatial distribution and deformation characteristics of active landslides in the UJSR.

## 7. Conclusions

Landslide identification was carried out using stacking-InSAR technology, optical satellite remote sensing images and UAV data. SBAS-InSAR was employed to obtain time-series deformation curves of samples to illustrate the deformation characteristics of active landslides. A total of 246 active landslides were identified in the UJSR through integrated remote sensing techniques and field surveys. The active landslides were concentrated in three zones, with a total of 207 landslides distributed within these three zones. Thirty-one landslides in the study area displayed large deformations and were chosen as the study samples. Based on SBAS-InSAR technology, time-series deformation curves of landslides with intense deformation zones between 2018 and 2022 were collected. The curve morphology enabled us to classify the landslide samples into four categories. Six linear-type landslides, three upward concave-type landslides, 10 downward concave-type landslides, and 12 stepped-type landslides were identified.

**Author Contributions:** Conceptualization, W.L. and B.C.; methodology, S.Z. and B.C.; software, B.C. and S.Z.; validation, H.L. and Z.L.; formal analysis, Y.S. and Z.L.; investigation, B.C. and S.Z.; resources, P.L. and X.C.; data curation, S.Z.; writing—original draft preparation, S.Z. and B.C.; writing—review and editing, W.L.; visualization, S.Z. and B.C.; supervision, W.L.; project administration, W.L.; funding acquisition, P.L. All authors have read and agreed to the published version of the manuscript.

**Funding:** This research was funded by National Key Research and Development Program of China (2021YFC3000401), the National Key Research and the National Natural Science Foundation of China (Grant No. 41941019), the Key Research and Development Program of Sichuan Province (Grant No. 2023YFS0435), the Yangtze River Joint Research Phase II Program (Grant No. 2022-LHYJ-02-0201), the China Power Construction Group Research Project (Grant No. DJ-ZDXM-2020-3), and the State Key Laboratory of Geohazard Prevention and Geoenvironment Protection Independent Research Project (Grant No. SKLGP2022Z007).

**Data Availability Statement:** Data are contained within the article.

**Acknowledgments:** Thanks to ESA for providing Sentinel-1A images. Thanks to Professor Xing Zhu of Chengdu University of Technology for providing the GNSS data of the Xiongba landslide.

**Conflicts of Interest:** Pengfei Li was employed by the Guiyang Engineering Corporation Limited of Power China. Xiong Cao was employed by Southwest Branch of China Petroleum Engineering and Construction Co., Ltd. The remaining authors declare that the research was conducted in the absence of any commercial or financial relationships that could be construed as a potential conflict of interest.

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
