# Peer review of "Analysis of the Spatial Distribution and Deformation Types of Active Landslides in the Upper Jinsha River, China, Using Integrated Remote Sensing Technologies"

_remotesensing, doi:10.3390/rs16010100_

Round 1
Reviewer 1 Report
Comments and Suggestions for Authors
The manuscript presented a comprehensive analysis of landslide detection and deformation characteristics in the upper Jinsha River, China. I appreciate the authors applied an integrated approach combining SBAS-InSAR, optical satellite remote sensing images and UAV to analyze the deformation mechanisms of active landslides in the upper Jinsha River. This research can positively contribute to the field of landslide detection and risk assessment, which is of interest to the readers of "Remote Sensing”. The manuscript reads well and it was well structured, but there are still several major issues that the authors have to consider. The detailed comments are listed as follows:
1. The Introduction section: lines 34-36 and lines 39-40 need references.
2. Figure 1: The legend is somehow confusing and needs further revision. And the sources of data such as rainfall, geology, and earthquake need to be introduced in detail.
3. The description of the Step-type (lines 423-430) in the manuscript is insufficient, more detailed description should be added according to Figure 15.
4. In lines 485-488, the classification of landslides into four types - stable, accelerated, convergent and step - is inconsistent with the previous statement, categorizing them into linear-type, upwards concave-type, downwards concave-type and step-type.
5. The discussion section is relatively simple, and it is recommended to strengthen it
6. The figures do not correspond well with the text. For example:
-Lines 236-249: fail to provide a description of Figures 6b, 6d and 6f.
-Line 450: lacks a description of Figure 16d.
-Figures 1d and 1e are not cited in the text.
-Line 105:'Fig. 1a, Fig. 1b' should be 'Figs. 1a and 1b'.
Based on this, I recommend a minor revision applied to the manuscript before its acceptance for publication.
Comments on the Quality of English LanguagePlease carefully the grammatical errors and improve English expression throughout the manuscript.
Reviewer 2 Report
Comments and Suggestions for Authors
There have been many researches using the same methodology, the same dataset, and the same study area in this region. This manuscript is lack of novelty. In addition, there are many language problems which is difficult to understand.
Comments on the Quality of English LanguageThere are a lot of English grammar problems in the manuscript which is very difficult to understand.
Reviewer 3 Report
Comments and Suggestions for Authors
The detection and deformation analysis of active landslides are critical in landslide risk management. The manuscript reported a case study of active landslide mapping and deformation characteristics analysis in the upper Jinsha River, China. This research could make a positive contribution to landslide mapping. While the manuscript might interest the readers of RS, it requires substantial improvement before it can be considered worthy of publication. The Reviewer is recommending that Revision is necessary. The detailed comments are listed as follows:
1. Lines 176-181: The description of optical remote sensing images is too simple, and it is necessary to supplement the reasons for selecting images of these specific dates.
2. Lines 319 to 329: lack of detailed description of the principle of SBAS-InSAR technology, it is inconsistent with the fact that the manuscript focuses on the time-series deformation characteristics of landslides.
3. The time series deformation curves in the manuscript are obtained by SBAS-InSAR technology, authors should add GNSS monitoring results to improve reliability of results.
4. Fig. 10a: lack of the label of Fig. f, and the label of Fig. h seems to be redundant.
5. The introduction and discussion sections need more supporting references.
6. Fig. 13 to Fig. 15: explain the meaning of the background color used in the line chart.
7. Some figures are suggested to be cited, such as Fig. 1d, Fig. 1e, Fig. 11, Table 2, etc.
Reviewer 4 Report
Comments and Suggestions for Authors
This manuscript presents a monitoring and type-identification landslide methodology to infer deformation values in active landslides. The research herein presented is certainly within the scope of Remote Sensing.
According to my observations, the topic of the manuscript is interesting and challenging. However, several points need to be addressed prior to its publication. I provide a list of comments below. I will be happy to review an updated version of the manuscript.
Comments:
1. The title is rather cumbersome and imprecise: the novelty of the manuscript is not highlighted and the reader is confused about the contents of itself (is it a numerical/theoretical/experimental manuscript?). Please, re-think the title.
2. The introduction should frame better the challenges addressed by this work. As it is, it jumps abruptly towards InSAR technique and landslide categories. I think the authors should first outline the existing monitoring methods to measure the landslides phenomena, i.e. GPS, borehole inclinometer [1], time-domain reflectometry (TDR) [2], optical fiber sensing technology [3] or RGB-D sensors to measure the surface deformation [4]. Only after describing these monitoring methods the authors can flow towards the definition of computational units and landslides categories.
3. Sections 2.2 and 2.3 need to include the spatial resolution per pixel of the satellite images and of the aerial images. Also, the height at which the UAV flight was carried out needs to be indicated.
4. Some of the landslides studied here seem to be deep-seated. One would therefore welcome a justification/discussion on the use of conditioning factors such as land use and NDVI.
5. Results Figure 13-15. How the accuracy in the deformation values estimation is related with landslide characteristics (velocity, soil depth, porosity, water content)?.
6. Discussion. The authors seem to approach the problem purely from a monitoring perspective (satellites and UAVs), which, while commendable, fails to grasp the complex and multifaceted nature of landslide phenomena. Landslide displacement is controlled by a host of factors including, but not limited to, lithology, stratigraphy, soil properties, landslide morphology and dynamics etc.. The current version of the paper does not adequately consider or integrate these factors in the analysis and in the discussion. The lack of a more comprehensive examination of the geological and geomorphological factors controlling landslides significantly weakens its overall impact.
7. Discussion. Following with my previous comment. All through the manuscript the deformation value is evaluated, but, why is it important for the Upper Jinsha River catchment management?. For instance, you should link it with additional sediment supply to the river reach. This is an important point in the management of large catchments, as it is your case study, and it is partially mentioned in the introduction. Please, see [6] and add some contextualization in your discussion.
Bibliography
[1] Zhang Y, Tang H, Li C, Lu G, Cai Y, Zhang J, Tan F. Design and Testing of a Flexible Inclinometer Probe for Model Tests of Landslide Deep Displacement Measurement. Sensors 2018, 18, 224.
[2] Su, M.-B.; Chen, I.-H.; Liao, C.-H. Using TDR cables and GPS for landslide monitoring in high mountain area. J. Geotech. 669 Geoenviron. Eng. 2009, 135, 1113–1121.
[3] Zhu HH, Shi B, Zhang CC. FBG-Based Monitoring of Geohazards: Current Status and Trends. Sensors 2017, 17, 452.
[5] C. Juez, D. Caviedes-Voullième, J. Murillo and P. García-Navarro. 2D dry granular free-surface transient flow over complex topography with obstacles. Part II: Numerical predictions of fluid structures and benchmarking. Computers & Geosciences, Volume 73, 2014.
[6] Intraseasonal‐to‐Interannual Analysis of Discharge and Suspended Sediment Concentration Time‐Series of the Upper Changjiang (Yangtze River). C. Juez et al. Water Resources Research. 2021.
Comments on the Quality of English LanguageThe English needs to be polished in some parts of the text.
Round 2
Reviewer 4 Report
Comments and Suggestions for Authors
The authors addressed most of my queries. In my second comment I proposed the authors to outline existing monitoring methods to measure landslides. They included most of the suggested techniques but they missed the RGB-D sensors [4]. Please, include this technique just before sending the manuscript for production. Anyway, the manuscript is in good shape and deserves to be accepted. I don't need to see it again.
